# Predictors of prior HIV testing and acceptance of a community-based HIV test offer among male bar patrons in northern Tanzania

Deng B. Madut[1,2]*, Preeti Manavalan[3], Antipas Mtalo[4], Timothy Peter[4], Jan Ostermann[5], Bernard Njau[6], Nathan M. Thielman[1,2]

1 Department of Medicine, Duke University, Durham, North Carolina, United States of America, 2 Duke Global Health Institute, Durham, North Carolina, United States of America, 3 Department of Medicine, University of Florida, Gainesville, Florida, United States of America, 4 Kilimanjaro Christian Medical Centre, Moshi, Tanzania, 5 Department of Health Services Policy and Management, University of South Carolina, Columbia, South Carolina, United States of America, 6 Kilimanjaro Christian Medical University College, Moshi, Tanzania

* deng.madut@duke.edu

**Data Availability Statement:** All relevant data are within the manuscript and its Supporting Information files.

## Abstract

Community-based HIV testing offers an alternative approach to encourage HIV testing among men in sub-Saharan Africa. In this study, we evaluated a community-based HIV testing strategy targeting male bar patrons in northern Tanzania to assess factors predictive of prior HIV testing and factors predictive of accepting a real-time HIV test offer. Participants completed a detailed survey and were offered HIV testing upon survey completion. Poisson regression was used to identify prevalence ratios for the association between potential predictors and prior HIV testing or real-time testing uptake. Of 359 participants analyzed, the median age was 41 (range 19–82) years, 257 (71.6%) reported a previous HIV test, and 321 (89.4%) accepted the real-time testing offer. Factors associated with previous testing for HIV (adjusted prevalence ratio [aPR], 95% CI) were wealth scores in the upper-middle quartile (1.25, 1.03–1.52) or upper quartile (1.35, 1.12–1.62) and HIV knowledge (1.04, 1.01–1.07). Factors that predicted real-time testing uptake were lower scores on the Gender-Equitable Men scale (0.99, 0.98–0.99), never testing for HIV (1.16, 1.03–1.31), and testing for HIV > 12 months prior (1.18, 1.06–1.31). We show that individual-level factors that influence the testing-seeking behaviors of men are not likely to impact their acceptance of an HIV offer.

## Introduction

HIV testing is the critical first step towards accessing HIV treatment services; yet, testing rates remain suboptimal across many settings in sub-Saharan Africa (SSA), particularly among men [1]. The relatively low uptake of HIV testing among men in SSA has been termed the HIV 'blind spot' and is increasingly recognized as a contributor to preventable morbidity and

**Funding:** This research was supported by the US National Institutes of Health (NIH) Fogarty International Center (grant number: D43TW009337, awarded to DBM) and the US NIH Ruth L. Kirschstein National Research Service Award (NRSA) (grant number: 5T32AI007392, awarded to DBM). The funders had no role in study design, data collection, analysis, decision to publish, or preparation of the manuscript.

**Competing interests:** The authors have declared that no competing interests exist.

mortality in the region [2]. To address this 'blind spot', a deeper understanding of barriers and facilitators of HIV testing among men is needed.

Existing evidence suggests that men are often reluctant to present to traditional health facilities for HIV testing [3, 4]. Factors consistently found to characterize men who report never testing for HIV include younger age, low education attainment, poor HIV knowledge, and stigmatizing views of HIV [2, 5–7]. There is also increasing recognition that societal-level constructs such as masculine ideals that emphasize strength and self-reliance represent barriers to men engaging in health-seeking behaviors [8]. At the facility level, barriers to testing include confidentiality concerns and the perception that clinics are female spaces [9]. Finally, the gender gap in HIV testing exists in part because antenatal care and other reproductive health services provide an entry point for women to access HIV care [10]. In contrast, men have fewer opportunities to interact and engage with the healthcare system.

To address the challenges men face in accessing HIV testing services across SSA, novel testing strategies have been implemented, with some showing encouraging results. Noticeably, community-based HIV testing strategies show promise in overcoming many of the structural and institutional barriers that tacitly exclude men from accessing existing HIV testing services [11]. However, the extent to which community-based testing can overcome individual-level characteristics that limit men's test-seeking behaviors remains uncertain. To this end, we previously conducted a study to evaluate if targeting male bar patrons in northern Tanzania was an efficient strategy for identifying undiagnosed men living with HIV [12]. We found that bars in northern Tanzania are patronized by men at increased risk for HIV and thus serve as opportune settings for targeted HIV testing. In the present study, we conducted a secondary analysis of these data and first described the individual-level factors associated with prior testing for HIV among male bar patrons. We then evaluated if these factors are also associated with HIV testing acceptance in the context of our study.

## Methods

### Setting

Our study was conducted from 6 December 2018 through 31 May 2019 in the town of Boma Ng'ombe, henceforth referred to as Boma, which is located in the Hai District of the Kilimanjaro Region of Tanzania. Boma lies on a major highway connecting the Kilimanjaro Region to the Arusha Region and has a population of approximately 17,000 persons. HIV prevalence among adults aged 15 years or older in the Kilimanjaro Region is estimated at 2.6% with a prevalence of 2.0% among men and 3.1% among women [13].

### Bar enrollment and sampling

Detailed procedures regarding bar enrollment and sampling have been previously described [12]. Briefly, a bar was defined as an establishment that sells alcohol and provides seating for the consumption of alcohol. All bars in Boma were eligible for enrollment, and a study team member visited each bar and requested permission from the bar owner to recruit male patrons. The days and times to visit bars for participant recruitment were randomized and adapted from previously described venue sampling methods [14]. Each month, 16 to 20 bars were randomly selected to recruit bar patrons. Per self-report from bar owners, customer traffic in bars was highest after 4 p.m. on most days. Out of safety considerations for study recruiters, participant recruitment at bars stopped after 8 p.m. Thus, participant recruitment from bars occurred from 4 p.m. through 8 p.m.

## Participant recruitment

All males aged 18 years or older entering selected bars were approached except for individuals with signs of intoxication such as slurred speech or disinhibition. The research assistant provided each eligible male with a study recruitment card that was uniquely numbered, dated, and valid for one month. This card invited patrons to report to our study office, located in Boma, on a day different than the recruitment day. Our study office was a standalone building that was readily accessible to community members and was not associated with any facility-based HIV testing centers. Office hours for enrollment were Mondays, Wednesdays, Thursdays, and Saturdays from 9 a.m. to 5 p.m. Individuals presenting to the office without a recruitment card were ineligible to participate. All participants were informed at the time of recruitment that they would be reimbursed 5,000 Tanzanian shillings (TSh), approximately 2.17 US Dollars in 2019 currency, for participation in the study.

## Survey

Eligible males presenting to the study office were offered enrollment into the study. After obtaining informed consent, participants underwent a survey administered in Kiswahili by Tanzanian research assistants using Samsung Galaxy Tab A tablets (Samsung, Seoul, South Korea). The survey was designed using Open Data Kit version 1.12.2 (available online at https://opendatakit.org/). Basic sociodemographic information, including age, marital status, and the highest level of education attained, was recorded. A wealth score was derived using principal component analysis from the following individual and household characteristics: educational attainment, quality of water supply, quality of toilet, quality of floor, number of rooms in the household, ownership of any low-cost household items such as a table or chair, ownership of any expensive household items such as a washer/dryer, computer, or air conditioner, electricity in the household, television ownership, refrigerator ownership, phone ownership, car ownership, and bicycle ownership. Alcohol use was measured using the Alcohol Use and Disorder Identification Test (AUDIT), which has been validated in Tanzania [15, 16]. Participants were asked about sexual activity in the last 12 months. Sexually active participants were considered to have concurrent sexual partners if they reported an ongoing sexual relationship with at least 2 partners. Attitudes towards gender norms were evaluated using the 24-item Gender Equitable Men (GEM) scale [17, 18]. This scale has a series of statements aimed at understanding men's views on the roles and behaviors of men and women. Scores range from 24 to 72 with higher scores reflecting higher support for gender equity. HIV knowledge was evaluated using the 18-item HIV Knowledge Questionnaire (HIV-KQ-18) [19]. HIV stigma was evaluated using the 9-item AIDS-related Stigma scale [20]. Because the majority of participants reported no stigmatizing attitudes, this variable was dichotomized as no stigma vs any stigma. Lastly, participants were asked about the timing of their last HIV test. Prior HIV testing history was categorized as '≤ 12 months', '>12 months', and 'Never tested'.

## HIV testing

During the study consenting process, all participants were offered HIV testing but were also made aware that HIV testing was not required to participate in the study and that refusal of testing would not affect study compensation. HIV testing was performed by a research assistant trained in voluntary counseling and testing. All testing was accompanied by pre- and post-test counseling and was performed on fingerstick samples using the SD-Bioline HIV-1/HIV-2 3.0 test (Standard Diagnostics Inc, Kyonggi-do, South Korea) followed by the Unigold Rapid HIV test (Trinity Biotech, Bray, Ireland) for confirmation of a positive SD-Bioline test. In the event of a positive test, participants were referred to a facility-based treatment center of

their choice. Participants received follow-up phone calls from a research assistant to encourage care engagement. We did not track care engagement or long-term outcomes for these participants.

### Statistical analysis

All analyses were performed using STATA version 16.0 (StataCorp, College Station, TX). Participants with known HIV infection at the time of enrollment were excluded from the analysis (Fig 1). Continuous variables were expressed using medians and ranges. Categorical variables were expressed as frequencies. Poisson regression models with robust variance were constructed to estimate unadjusted and adjusted prevalence ratios (aPRs) and 95% confidence intervals (CIs) for associations between individual-level predictors and dependent variables

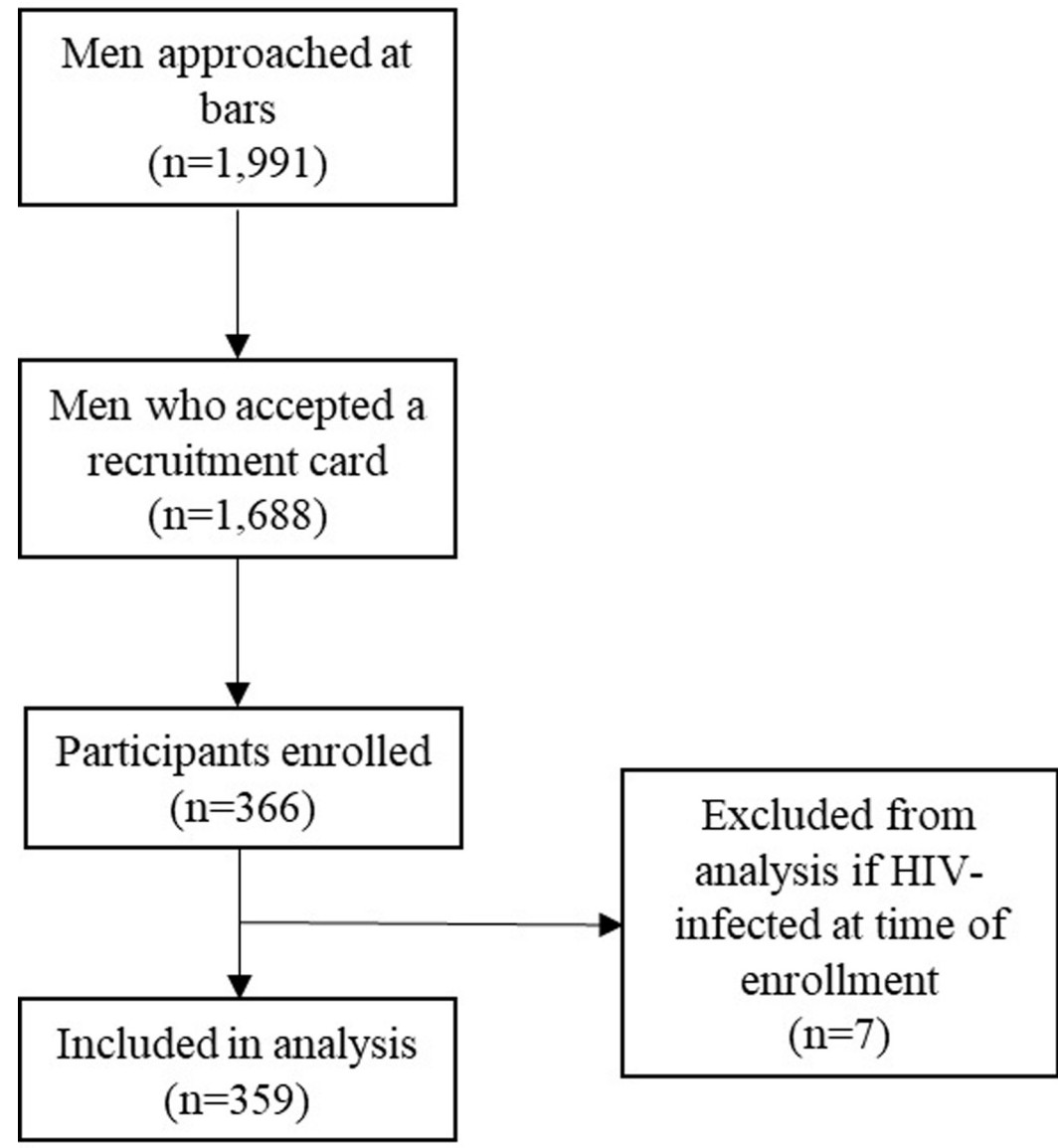

**Fig 1. Flow diagram of participants selected in a study of male bar patrons in northern Tanzania (2018–2019).**

(prior testing and HIV testing acceptance) [21]. All *P*-values are two-sided and a *P* <0.05 was considered statistically significant.

### Ethical considerations

Written consent was obtained from all participants. All consent forms were stored in locked cabinets in a room with controlled entry. Ethics approval was obtained from the Research Ethics Committee of Kilimanjaro Christian Medical Centre (No. 2250), the Ethics Coordinating Committee of the Tanzanian National Institute for Medical Research (NIMR/HQ/R.8c/Vol.1/ 1192), and the Institutional Review Board of Duke University (Pro00083626).

## Results

### Participant characteristics

We analyzed data from 359 participants recruited from 72 different bars in Boma (Fig 1). Of participants, the median age was 41 (range 19–82) years, 68 (18.9%) were never married, 67 (18.7%) reported secondary school or higher education, 205 (57.1%) reported high risk alcohol use (AUDIT ≥ 8), and 45 (12.5%) reported concurrent sexual partners. Participant characteristics are provided in Table 1.

### HIV testing behaviors

Detailed HIV testing results from our study have been previously published [12]. Of participants, 257 (71.6%) reported a previous HIV test and 102 (28.4%) reported no previous HIV testing. Among participants who reported a previous HIV test, 99 (38.5%) reported that their last test was ≤ 12 months prior and 156 (61.5%) reported testing > 12 months prior. HIV testing was offered to all participants at the time of study enrollment, and 321 (89.4%) accepted HIV testing while 38 (10.6%) declined. Among those who accepted testing, 95 (29.6%) had not previously tested and 17 (5.4%) were newly diagnosed with HIV. We detected 1 (1.3%) infection among participants who reported testing ≤ 12 months prior, 10 (6.8%) among those who tested > 12 months prior, and 6 (6.3%) among never-testers.

When participants without a previous HIV test were asked the primary reason for never testing, 45 (44.1%) reported that they considered themselves "Not at risk for HIV". This was the most common reason for never testing followed by "Could not leave work to get tested" which was reported by 20 (19.6%) participants. When participants were asked the primary reason they declined our HIV test offer, 15 (39.5%) participants responded "I know my status/ Recent HIV test". This was the most common reason for declining the study test offer followed by "Not ready to test" which was reported by 7 (18.4%) participants. Participants' reasons for never testing and declining our HIV test offer are presented in Table 2.

**Factors associated with HIV testing behaviors.** Bivariable regression results are presented in Table 3. In the final multivariable models (adjusted prevalence ratio [aPR], 95% confidence interval [CI]), factors associated with previous HIV testing were wealth score in the upper middle quartile (1.25, 1.03–1.52), wealth score in the top quartile (1.35, 1.12–1.62), and HIV knowledge (1.04, 1.01–1.07). In comparison, factors predictive of accepting the study's HIV test offer were lower GEMS score (0.99, 0.98–0.99), testing for HIV > 12 months prior, (1.18, 1.06–1.31), and never testing for HIV (1.16, 1.03–1.31).

## Discussion

We have described predictors of previous HIV testing and that of accepting an HIV test in the context of a community-based testing strategy targeting male bar patrons in northern

**Table 1. Characteristics of men recruited from bars in northern Tanzania (2018–2019).**

| Variables | Total | Ever tested | | Response to HIV test offer | |
|---|---|---|---|---|---|
| | (N = 359) | Yes (N = 257) | No (N = 102) | Accepted (N = 321) | Declined (N = 38) |
| **Demographic factors** | | | | | |
| Age in years, median (range) | 41 (19–82) | 41 (19–82) | 42 (19–77) | 41 (19–80) | 42 (19–82) |
| Education, n (%) | | | | | |
| Primary or less | 292 (81.3) | 201 (78.2) | 91 (89.2) | 273 (85.1) | 19 (50.0) |
| Secondary or above | 67 (18.7) | 56 (21.8) | 11 (10.8) | 48 (14.9) | 19 (50.0) |
| Marital status, n (%) | | | | | |
| Married, divorced, or widowed | 291 (81.1) | 213 (82.9) | 78 (76.5) | 258 (80.4) | 33 (86.8) |
| Never married | 68 (18.9) | 44 (17.1) | 24 (23.5) | 63 (19.6) | 5 (13.2) |
| Wealth score, n (%) | | | | | |
| Bottom quartile | 95 (26.5) | 57 (22.2) | 38 (37.3) | 89 (27.7) | 6 (15.8) |
| Lower middle quartile | 81 (22.6) | 52 (20.2) | 29 (28.4) | 72 (22.4) | 9 (23.7) |
| Upper middle quartile | 92 (25.6) | 71 (27.6) | 21 (20.6) | 85 (26.5) | 7 (18.4) |
| Top quartile | 91 (25.3) | 77 (30.0) | 14 (13.7) | 75 (23.4) | 16 (42.1) |
| **Risk behaviors** | | | | | |
| AUDIT score, n (%) | | | | | |
| Low-risk use | 154 (42.9) | 108 (42.0) | 46 (45.1) | 134 (41.7) | 20 (52.6) |
| High-risk use | 205 (57.1) | 149 (58.0) | 56 (54.9) | 187 (58.3) | 18 (47.4) |
| Concurrent sexual partners, n (%) | 45 (12.5) | 37 (14.4) | 8 (7.8) | 42 (13.1) | 3 (7.9) |
| **Sociocultural factors** | | | | | |
| GEM scale, median (range) | 58 (34–71) | 58 (34–71) | 56 (38–70) | 58 (34–70) | 61 (44–71) |
| HIV knowledge score, median (range) | 14 (3–18) | 14 (5–18) | 13 (3–17) | 14 (3–17) | 14 (9–18) |
| Stigma score, n (%) | | | | | |
| No stigma | 154 (42.9) | 113 (44.3) | 40 (39.2) | 137 (42.7) | 17 (44.7) |
| Any stigma | 205 (57.1) | 143 (55.6) | 62 (60.8) | 184 (57.3) | 21 (55.3) |
| **HIV testing** | | | | | |
| HIV testing history, n (%) | | | | | |
| Last HIV test ≤ 12 months | 99 (27.6) | 99 (38.5) | - | 78 (24.3) | 21 (55.3) |
| Last HIV test > 12 months | 158 (44.0) | 158 (61.5) | - | 148 (46.1) | 10 (26.3) |
| Never testers | 102 (28.4) | - | 102 (100) | 95 (29.6) | 7 (18.4) |

Abbreviations: AUDIT, Alcohol Use Disorders Identification Test; GEM, Gender Equitable Men

Tanzania. Compared to men who reported never testing for HIV, men with a previous HIV testing history were in the upper wealth quartiles and displayed higher HIV knowledge. Factors that predicted real-time uptake of an HIV testing offer differed. Specifically, men who accepted our testing offer reported lower support for gender equity and were more likely to have never tested or had a distant testing history; wealth score quartile and HIV knowledge were not predictive.

Our finding that previous testing was associated with wealth and HIV knowledge is supported by prior studies. Multiple studies have found that both men and women of lower socioeconomic position face unique barriers to accessing HIV testing services across SSA [5, 7, 22]. However, the expansion of HIV testing within reproductive health services has helped to reduce socioeconomic obstacles faced by women [22]. Studies have also described an association between HIV knowledge and HIV testing among men [2, 5]. Increasing men's HIV knowledge could be achieved through simple education programs although it remains unclear to what extent mass HIV education campaigns result in subsequent engagement in HIV

**Table 2. Self-reported reasons for never testing for HIV and declining an HIV test offer among male bar patrons in northern Tanzania, (2018–2019).**

| What is the main reason you have never tested for HIV? (N = 102) | N | (%) |
|---|---|---|
| Not at risk for HIV | 45 | (44.1) |
| Can't leave work to get tested | 20 | (19.6) |
| Nervous to get the results | 13 | (12.6) |
| Never thought about getting an HIV test | 11 | (10.6) |
| Didn't know where to get tested | 6 | (5.8) |
| Other reason | 7 | (6.9) |
| **What is your reason for declining our HIV test offer? (N = 38)** | **N** | **(%)** |
| I know my status/Recent test | 15 | (39.5) |
| Not ready to test | 7 | (18.4) |
| Not at risk | 5 | (13.2) |
| Prefer to test with a partner | 5 | (13.2) |
| Other reason | 6 | (15.8) |

prevention services [23]. While other studies have found that older men, married or previously married men, and those not hold stigmatizing views of HIV are more likely to report prior HIV testing, we observed no association between these variables and prior testing in our study testing [2, 7, 24].

Our HIV test offer was made in the context of a research study; thus, evaluation of our findings outside of a formal research setting is needed. Despite this limitation, several important insights can be extracted from our study results. First, testing acceptance was high with nearly 90% of participants accepting our test offer. Notably, participants were aware that study enrollment and reimbursements were not contingent on testing acceptance. Second, testing acceptance was highest for never-testers and for men who reported their last HIV test as more than 12 months prior. We also detected a higher proportion of HIV infection among these men [12]. These results suggest that targeting men who have never tested or those with a distant HIV test history could optimize the efficiency of our testing strategy. Finally, wealth status and HIV knowledge were associated with prior testing, but these variables were not associated with the acceptance of our test offer. Other studies have found similar results suggesting that individual-level characteristics that influence HIV test seeking are not necessarily the same as those influencing a man's decision to accept testing [7, 25].

One finding that warrants further discussion was that men who reported lower support for gender equity were more likely to accept our testing offer. The role that gender norms play in the testing decisions of men is an area of active research. Some studies suggest that traditional masculine ideals such as those that emphasize strength and independence are often viewed as non-conforming with HIV test-seeking behaviors [8, 26]. However, others have found that these same masculine ideals facilitate HIV testing. Taken together, our results provide further evidence that the relationship between gender norms and HIV testing is complex and likely varies across different settings. Carefully exploring this relationship across different contexts could result in useful policy recommendations.

Our study has several limitations. First, data collected from participants largely relied on self-report. Self-report is subject to recall and social desirability biases [27]. Second, research findings from community-based testing strategies are likely to vary across different contexts; thus, caution should be exercised in generalizing our results to other settings across Tanzania and SSA. Third, the extent of selection bias in our study cannot be evaluated. Specifically, men

**Table 3. Predictors of HIV testing behaviors among men recruited from bars in northern Tanzania (2018–2019).**

| Variables | Predictors of previous testing | | Predictors of HIV test uptake | |
|---|---|---|---|---|
| | Unadjusted PR (95% CI) | Adjusted PR (95% CI) | Unadjusted PR (95% CI) | Adjusted PR (95% CI) |
| **Demographic factors** | | | | |
| Age in years | 1.00 (0.99–1.01) | 1.00 (0.99–1.01) | 0.99 (0.99–1.00) | 0.99 (0.99–1.00) |
| Marital status | | | | |
| Married, divorced, or widowed | Reference | Reference | Reference | Reference |
| Never married | 0.88 (0.73–1.07) | 0.86 (0.71–1.05) | 1.04 (0.97–1.13) | 1.02 (0.92–1.12) |
| Wealth score | | | | |
| Bottom quartile | Reference | Reference | Reference | Reference |
| Lower middle quartile | 1.07 (0.85–1.35) | 1.06 (0.84–1.32) | 0.94 (0.86–1.04) | 0.96 (0.87–1.05) |
| Upper middle quartile | 1.29 (1.05–1.57)* | 1.25 (1.03–1.52)* | 0.99 (0.91–1.07) | 1.01 (0.94–1.10) |
| Top quartile | 1.41 (1.17–1.70)* | 1.35 (1.12–1.62)* | 0.88 (0.89–0.98)* | 0.92 (0.83–1.02) |
| **Risk behaviors** | | | | |
| AUDIT score | | | | |
| Low-risk use | Reference | Reference | Reference | Reference |
| High-risk use | 1.04 (0.91–1.18) | 1.03 (0.90–1.17) | 1.05 (0.97–1.13) | 1.04 (0.97–1.12) |
| Concurrent sexual partners | 1.17 (1.01–1.37)* | 1.17 (0.99–1.37) | 1.05 (0.96–1.15) | 1.06 (0.96–1.16) |
| **Knowledge and psychosocial factors** | | | | |
| GEM scale | 1.01 (0.99–1.02) | 1.00 (0.99–1.37) | 0.99 (0.98–0.99)* | 0.99 (0.98–0.99)* |
| HIV knowledge score | 1.05 (1.02–1.08)* | 1.04 (1.01–1.07)* | 0.99 (0.80–1.15) | 0.99 (0.99–1.01) |
| Stigma score | | | | |
| No stigma | Reference | Reference | Reference | Reference |
| Any stigma | 0.94 (0.83–1.07) | 0.99 (0.88–1.14) | 1.01 (0.94–1.08) | 0.98 (0.92–1.05) |
| **HIV testing behaviors** | | | | |
| HIV testing history | | | | |
| Last HIV test ≤ 12 months | - | - | Reference | Reference |
| Last HIV test > 12 months | - | - | 1.19 (1.06–1.33)* | 1.18 (1.06–1.31) |
| Never testers | - | - | 1.18 (1.05–1.33)* | 1.16 (1.03–1.31)* |

Abbreviations: PR, prevalence ratio; AUDIT, Alcohol Use Disorders Identification Test; GEM, Gender Equitable Men

*$P < 0.05$

who enrolled in our study may differ from the larger population of men attending bars in northern Tanzania [28, 29].

## Conclusion

In summary, we described a community-based HIV testing strategy targeting male bar patrons in northern Tanzania and assessed the factors predictive of both past and real-time HIV testing uptake. Individual-level factors that predicted previous HIV testing were not associated with the uptake of our real-time HIV test offer. Taken together, our results provide further evidence that men are willing to test for HIV, and efforts to expand the reach of testing may improve testing coverage. Because men across SSA report a preference to test outside of traditional health systems, the community settings where they congregate should be leveraged to deliver HIV testing services.

## Supporting information

**S1 Data. Full study dataset.**
(XLS)

**S1 File. STROBE checklist.**
(DOCX)

**S2 File. Survey instrument.**
(XLSX)

**S1 Text. Inclusivity in global research.**
(DOCX)

## Author Contributions

**Conceptualization:** Deng B. Madut, Bernard Njau, Nathan M. Thielman.

**Data curation:** Deng B. Madut.

**Formal analysis:** Deng B. Madut.

**Funding acquisition:** Deng B. Madut, Nathan M. Thielman.

**Investigation:** Deng B. Madut.

**Methodology:** Deng B. Madut.

**Project administration:** Deng B. Madut, Antipas Mtalo, Timothy Peter.

**Supervision:** Bernard Njau, Nathan M. Thielman.

**Writing – original draft:** Deng B. Madut, Preeti Manavalan, Antipas Mtalo, Timothy Peter, Jan Ostermann, Bernard Njau, Nathan M. Thielman.

**Writing – review & editing:** Deng B. Madut, Preeti Manavalan, Antipas Mtalo, Timothy Peter, Jan Ostermann, Bernard Njau, Nathan M. Thielman.

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
