## [Decision Letter · Decision Letter 0]

4 Sep 2023

PGPH-D-23-01174

Predictors of prior HIV testing and acceptance of a community-based HIV test offer among male bar patrons in northern Tanzania

Dear Dr. Madut,

Thank you for submitting your manuscript to PLOS Global Public Health. After careful consideration, we feel that it has merit but does not fully meet PLOS Global Public Health’s publication criteria as it currently stands. Therefore, we invite you to submit a revised version of the manuscript that addresses the points raised during the review process.

Please see the comments from two reviewers below. We now invite you to consider each comment, and either rebut a concern or revise the manuscript accordingly.

Please also ensure that you complete the PLOS ONE Inclusivity in Global Research questionnaire when you resubmit: https://plos.org/wp-content/uploads/2023/07/Inclusivity-in-global-research-questionnaire.docx 

We look forward to receiving your revised manuscript.

Kind regards,

Hanna Landenmark

Staff Editor

Journal Requirements:

1. Please include a complete copy of PLOS’ questionnaire on inclusivity in global research in your revised manuscript. Our policy for research in this area aims to improve transparency in the reporting of research performed outside of researchers’ own country or community. The policy applies to researchers who have travelled to a different country to conduct research, research with Indigenous populations or their lands, and research on cultural artefacts. The questionnaire can also be requested at the journal’s discretion for any other submissions, even if these conditions are not met.  Please find more information on the policy and a link to download a blank copy of the questionnaire here: https://journals.plos.org/plosone/s/best-practices-in-research-reporting. Please upload a completed version of your questionnaire as Supporting Information when you resubmit your manuscript.

Additional Editor Comments (if provided):

Reviewers' comments:

Reviewer's Responses to Questions

**Comments to the Author**

1. Does this manuscript meet PLOS Global Public Health’s publication criteria? Is the manuscript technically sound, and do the data support the conclusions? The manuscript must describe methodologically and ethically rigorous research with conclusions that are appropriately drawn based on the data presented.

Reviewer #1: Yes

Reviewer #2: Yes

2. Has the statistical analysis been performed appropriately and rigorously?

Reviewer #1: Yes

Reviewer #2: No

3. Have the authors made all data underlying the findings in their manuscript fully available (please refer to the Data Availability Statement at the start of the manuscript PDF file)?

Reviewer #1: Yes

Reviewer #2: Yes

4. Is the manuscript presented in an intelligible fashion and written in standard English?

Reviewer #1: Yes

Reviewer #2: Yes

5. Review Comments to the Author

Reviewer #1: Thank you for the chance to review this paper on an intervention to increase access to community-based HIV testing in bars in a small town in northern Tanzania.

The US funded study looked at acceptability of different testing options in 359 men who completed a detailed survey and were then offered an HIV test, together with the factors associated with test uptake. The study also diagnosed 17 new infections in

The study strengths include looking into a broad range of factors including HIV knowledge, behaviour and views relating to gender norms.

This is an interesting and well written paper that could be published and I don’t have further comments or questions overall. I was however interested in the outcomes for the 17 people who tested HIV positive and whose lives will have been significantly changed from helping with a research survey.

It would be good to know about their pathways to care etc.

Reviewer #2: This paper assesses HIV testing uptake among male patrons in Tanzania. HIV testing uptake remains lower than that for women; thus, efforts to understand factors associated with HIV testing uptake among men are urgently needed. While this research is important, there are quite a number of areas that still need to be addressed to improve clarity.

Main comments

1. The authors write about factors associated with prior and current HIV testing uptake. However, it is not clear why the authors chose to concentrate on both prior and current HIV testing uptake instead of concentrating on one of the two areas. In my view, a focus on factors associated with current HIV testing uptake would suffice with no need to examine factors associated with prior HIV testing uptake. If the authors insist on including both areas, the authors should justify why it is crucial to examine both areas given the current trends in HIV testing behaviors among men.

2. This manuscript focuses on community-based HIV testing. However, in the entire manuscript, I did not see any description of community-based HIV testing. What I saw was that men were approached in the bars and they were asked to come to the research office for eligibility screening. If eligible, men were interviewed at the same place of screening. So, where is the community-based HIV testing referred to, which forms the main gist of the paper? In response, I would recommend that the authors provide a clear description of the community-based HIV testing offered, and how the issue of recruiting men from bars and interviewing them at the research office fits into this arrangement.

3. On page 11, the authors write, “Among those who accepted testing, 17 (5.4%) were found to be HIV-infected. We detected 1 (1.3%) infection among participants who reported testing ≤ 12 months prior, 10 (6.8%) among those who tested > 12 months prior, and 6 (6.3%) among never-testers”. I could not trace where the authors picked this information from. If it was published in a previous paper, then only reference can be made to that paper but no results should be reproduced as if these findings were part of the current analysis.

Minor comments

1. The authors should describe the context in the bars that led the study team to consider them as study sites. This information would help the readers to appreciate why the study team opted to recruit men from bars rather than from other places.

2. The authors indicate that all the men at the bars were approached and invited for interviews with the exception of those that were intoxicated. However, interviews were conducted on a day different from the recruitment day. If this was the case, then, why would they not invite every man at the bars to come to the screening site – since they would not be intoxicated at the time of interview? I would understand the issue of leaving out intoxicated men if the interviews were to be conducted at the bar, immediately after being selected.

3. Although the authors refer to “predictors of prior HIV testing”, I did not see any description of what constituted “prior HIV testing”. How was prior HIV testing defined and measured? There was no indication that any data on prior HIV testing were collected.

4. Please include the study approval numbers from the Research and Ethics Committees that approved the study.

5. On page 10, the authors present the “baseline characteristics of the cohort”. I could not understand what “baseline” visit that the authors referred to since there was no prior mention of baseline or follow-up visits before getting to page 10; and there is also no mention of any cohort that was followed up as part of the study. If this analysis is based on data collected as part of a population-based cohort, then, there should be a description of the cohort from which data were drawn as part of the ‘Methods’ section. But even then, I would not refer to ‘baseline’ unless there is a particular reason for doing so.

6. In Table 1, the authors refer to ‘Response to HIV testing offer’ as a correlate under the “HIV testing” characteristic but also indicate a possible cross-tabulation with ‘Response to HIV test offer’. In one way or the other, there is duplication here that can be avoided. To understand what I mean, the authors can re-examine how “response to HIV testing offer” (coded as “declined/accepted”) was cross-tabulated with “response to HIV test offer” (coded as “declined/accepted”). One of the two aspects would have to be dropped.

7. The word “HIV infected” should be replaced. It is no longer in use as it is considered stigmatizing.

8. I did not see any attempt to characterize the men interviewed as part of the study as “bar patrons”, other than the fact that this is mentioned in the paper. How can we be sure that the men interviewed were bar patrons? This is important given that the findings reported in the paper can also be reported from any other paper that targeted men in the general population. Also, any failure to confirm that the study participants were indeed male patrons in a bar setting affects the generalizability of the findings. I think more information can be provided about the men that were recruited into the study, to characterize them as bar patrons, different from men picked from the general population.

9. Finally, in the discussion section, the bar setting should be one of the issues discussed. This should begin by identifying bars as places where risk behavior occurs, and then describing male patrons as an HIV-risk group. The discussion can then continue to focus on why recruiting men from such setting was necessary and the implications of the findings on research and HIV prevention efforts in similar settings.

6. PLOS authors have the option to publish the peer review history of their article (what does this mean?). If published, this will include your full peer review and any attached files.

**Do you want your identity to be public for this peer review?** For information about this choice, including consent withdrawal, please see our Privacy Policy.

Reviewer #1: No

Reviewer #2: No

---

## [Decision Letter · Decision Letter 1]

22 Dec 2023

PGPH-D-23-01174R1

Predictors of prior HIV testing and acceptance of a community-based HIV test offer among male bar patrons in northern Tanzania

Dear Dr. Deng,

Thank you for submitting your manuscript to PLOS Global Public Health. After careful consideration, we feel that it has merit but does not fully meet PLOS Global Public Health’s publication criteria as it currently stands. Therefore, we invite you to submit a revised version of the manuscript that addresses the points raised during the review process.

We look forward to receiving your revised manuscript.

Kind regards,

Joseph KB Matovu, Ph.D.

Guest Editor

Journal Requirements:

1. Please include a complete copy of PLOS’ questionnaire on inclusivity in global research in your revised manuscript. Our policy for research in this area aims to improve transparency in the reporting of research performed outside of researchers’ own country or community. The policy applies to researchers who have travelled to a different country to conduct research, research with Indigenous populations or their lands, and research on cultural artefacts. The questionnaire can also be requested at the journal’s discretion for any other submissions, even if these conditions are not met.  Please find more information on the policy and a link to download a blank copy of the questionnaire here: https://journals.plos.org/plosone/s/best-practices-in-research-reporting. Please upload a completed version of your questionnaire as Supporting Information when you resubmit your manuscript.

Additional Editor Comments (if provided):

Reviewers' comments:

Reviewer's Responses to Questions

**Comments to the Author**

1. If the authors have adequately addressed your comments raised in a previous round of review and you feel that this manuscript is now acceptable for publication, you may indicate that here to bypass the “Comments to the Author” section, enter your conflict of interest statement in the “Confidential to Editor” section, and submit your "Accept" recommendation.

Reviewer #3: (No Response)

Reviewer #4: All comments have been addressed

2. Does this manuscript meet PLOS Global Public Health’s publication criteria? Is the manuscript technically sound, and do the data support the conclusions? The manuscript must describe methodologically and ethically rigorous research with conclusions that are appropriately drawn based on the data presented.

Reviewer #3: Yes

Reviewer #4: Yes

3. Has the statistical analysis been performed appropriately and rigorously?

Reviewer #3: Yes

Reviewer #4: Yes

4. Have the authors made all data underlying the findings in their manuscript fully available (please refer to the Data Availability Statement at the start of the manuscript PDF file)?

Reviewer #3: Yes

Reviewer #4: Yes

5. Is the manuscript presented in an intelligible fashion and written in standard English?

Reviewer #3: Yes

Reviewer #4: Yes

6. Review Comments to the Author

Reviewer #3: Summary and Overall impression

This is a unique study that has explored a special area of the Community-based HIV counselling and testing, a model that has been of interest in research over the past 2 decades. The study has demonstrated an appreciable level of innovativeness in that it targeted and explored predicting factors of previous HIV testing and test acceptance among a special at-risk group, men. The authors have been able to nicely prove the hypothesis that community-based HIV testing could yield a better result than health facility-based VCT, a finding which has been clearly proven by earlier studies including K. Champenois et al. and C. Arlene et al. in 2009. It further demonstrated novelty in targeting male bar patrons, which was earlier suggested by S. Wilson et al. with a clear reproducible methodological approach with tight assurance of participants’ confidentiality whiles providing real time HIV testing to participants. It once more demonstrated the impact of HIV knowledge, wealth scores and gender norms on HIV testing as earlier elucidated by Musumari, Patou Masika, et al., K. Jean et al and L. Johnson respectively. Finally, it is very interesting to see all possible limitations of the study clearly outlined by the authors.

Major Issues

There were few minor numerical inconsistencies I have noticed which I think may need to be addressed for the sake uniformity.

1. For instance, in Line 182 on page 9, the 15 (39.4%) was a little different from that in the table 2 as 15 (39.5%) which were both rounded to one decimal place. Since the real ratio is 15/38=39.47, I think it would be better to keep it 39.5% to one decimal place to make the in-text figures consistent with the figures in the tables.

2. Same applies to Line 39 on page 2 under Abstract, where adjusted Prevalence Ratios and 95%CI ……HIV knowledge (1.05, 1.02-1.08), which was a little different from line 194 on page 10……HIV knowledge (1.04, 1.01-1.07).

Minor issues

There was no other issue regarding this manuscript to the best of my knowledge.

Reviewer #4: Predictors of prior HIV testing and acceptance of a community-based HIV test offer among male bar patrons in northern Tanzania

Thanks for giving me the opportunity to review this exciting manuscript. I see a marked improvement from the previous one. I have some minor concerns for the authors to consider.

Results

Table 3: I am wondering why the authors decided to combine married, divorced and widowed together. I am not sure if their willingness to take risk or risky behavior will be same. Maybe the authors could consider segregating this and use married as the reference.

Discussion

I suggest that this should rather be discussion and conclusion since the last paragraph concludes the study.

7. PLOS authors have the option to publish the peer review history of their article (what does this mean?). If published, this will include your full peer review and any attached files.

**Do you want your identity to be public for this peer review?** For information about this choice, including consent withdrawal, please see our Privacy Policy.

Reviewer #3: **Yes: **Dr Abraham Kwadzo Ahiakpa

Reviewer #4: **Yes: **Stephen Apanga

---

## [Decision Letter · Decision Letter 2]

30 Jan 2024

Predictors of prior HIV testing and acceptance of a community-based HIV test offer among male bar patrons in northern Tanzania

PGPH-D-23-01174R2

Dear Dr. Madut,

We are pleased to inform you that your manuscript 'Predictors of prior HIV testing and acceptance of a community-based HIV test offer among male bar patrons in northern Tanzania' has been provisionally accepted for publication in PLOS Global Public Health.

Best regards,

Joseph KB Matovu, Ph.D.

Guest Editor

Reviewer Comments (if any, and for reference):

Reviewer's Responses to Questions

**Comments to the Author**

1. If the authors have adequately addressed your comments raised in a previous round of review and you feel that this manuscript is now acceptable for publication, you may indicate that here to bypass the “Comments to the Author” section, enter your conflict of interest statement in the “Confidential to Editor” section, and submit your "Accept" recommendation.

Reviewer #4: All comments have been addressed

2. Does this manuscript meet PLOS Global Public Health’s publication criteria? Is the manuscript technically sound, and do the data support the conclusions? The manuscript must describe methodologically and ethically rigorous research with conclusions that are appropriately drawn based on the data presented.

Reviewer #4: Yes

3. Has the statistical analysis been performed appropriately and rigorously?

Reviewer #4: Yes

4. Have the authors made all data underlying the findings in their manuscript fully available (please refer to the Data Availability Statement at the start of the manuscript PDF file)?

Reviewer #4: (No Response)

5. Is the manuscript presented in an intelligible fashion and written in standard English?

Reviewer #4: Yes

6. Review Comments to the Author

Reviewer #4: Comments adequately addressed.

7. PLOS authors have the option to publish the peer review history of their article (what does this mean?). If published, this will include your full peer review and any attached files.

**Do you want your identity to be public for this peer review?** For information about this choice, including consent withdrawal, please see our Privacy Policy.

Reviewer #4: **Yes: **Stephen Apanga
